# Estimating the Impact of Digital Nomads' Sustainable Responsibility on Entrepreneurial Self-Efficacy

Inês Mourato [1], Álvaro Dias [1,2,*] and Leandro Pereira [1,2]

1   DMOGG, ISCTE—University Institute of Lisbon, 1649-026 Lisbon, Portugal
2   BRU-Business Research Unit, ISCTE—University Institute of Lisbon, 1649-026 Lisbon, Portugal
*   Correspondence: alvaro.dias@iscte-iul.pt

**Abstract:** Digital nomads live outside of the classical organizational borders and can be seen as 'contemporary entrepreneurs' who bring disruptive business models into different industries, giving value to different working cultures and different types of capital. Because they are operating out of their home country, their social responsibility as entrepreneurs may have different implications. This study aims to explore the outcomes of digital nomads' social responsibility in terms of self-efficacy and innovation. To test the hypothesis model, structural equation modeling (SEM) was used to analyze survey data. The results show that tourism firms should always have in mind their social responsibility levels to attract this target niche and should pay attention to the fact that they are mostly solo travelers, so they value the sense of community of a place and its legal, bureaucratic, and flexible terms of living. Findings also show that social and environmental concerns are more associated to social self-efficacy than to self-efficacy. The results complement existing research by helping tourism businesses and destination managers to understand the implications of the digital nomads' social responsibility.

**Keywords:** tourism entrepreneurship; digital nomads; self-efficacy; innovation





## 1. Introduction

The development of portable technologies and the liberalization of the airspace originated new lifestyles, such as digital nomadism. Companies are aware that workers are able to work from wherever they want to, enabling them to have a more dynamic way of working (Brown and O'Hara 2003). This way of living is facilitated by a combination of improved global access to information and information infrastructures, more flexible work arrangements, and the sense for adventure among the younger generation of knowledge workers (Dal Fiore et al. 2014). Digital nomads live outside of the classical organizational borders (Makimoto and Manners 1997) and can be seen as 'contemporary entrepreneurs' who bring disruptive business models into different industries (Vieira 2016), giving value to different working cultures and different types of capital (e.g., reputation, information, symbolic) (Nash et al. 2018).

There is already literature concerning the digital nomads' motivations and lifestyle. Thompson (2019) suggested that this kind of worker has an identity based on their lifestyle, and that is why they come together to the same conferences and retreats—to meet people alike and reinforce their individuality. The author also provides a critique on the privilege and inequality of this lifestyle, usually overlooked in the entrepreneurial literature. Hannonen (2020) studied the differentiating factors between digital nomadism and other lifestyle-led mobilities and mobile remote work. Additionally, the author tried to define aspects, such as, "the importance of labour productivity in digital nomadism, the state of international (semi) perpetual travel, downshifting, lifestyle-led bonding and communities and nomadicity of work" (Hannonen 2020, p. 17). Reichenberger (2018) illustrated that their professional and spatial freedom contributed to digital nomads' personal freedom by

creating a holistic lifestyle of opportunities for self-development and learning. Nash et al. (2021) focused their studies on the dynamic relationship between space, work, and technology, suggesting that labelling digital nomads as location-independent nomadic workers is a miscategorization. Mancinelli (2020) concluded that this type of traveler has a minimalist attitude toward property and consumption and gives importance to flexibility and entrepreneurialism. Finally, Chevtaeva and Denizci-Guillet (2021) examined the connection between digital nomads' personal lifestyles and perceptions of the value of coworking spaces during travel. This being said, the literature still needs further development, especially as underexplored theme related to the social and environmental responsibility of this type of traveler and their relationship and actions with the local ecosystem.

This study aims to understand the sustainable responsibility of the digital nomads to further conclude how they influence or could influence the local communities and how the tourism businesses can take an important step by understanding their needs and actions. By doing so, this study addresses Green's (2020) request for digital nomads' work sensemaking. It relates sustainable responsibility variables, such as social responsibility and environmental concern, with entrepreneurial variables, such as entrepreneurial self-efficacy and eco-innovation, having only digital nomads as the target population. Therefore, the current study aims to answer the following research question: what are the constraints of the social responsibility of digital nomads?

A quantitative method was used with the help of a snowballing technique, where several questionnaires were delivered (see Appendix A). This study proved to be useful for tourism businesses by allowing them to understand how to deal with digital nomads in a more practical way. Tourism firms should always have in mind their social responsibility levels in order to attract this target niche. On the other hand, by understanding that most of them are entrepreneurs, tourism should facilitate and foster entrepreneurial events, workshops, and activities. Finally, and most importantly, they should pay attention to the fact that they are mostly solo travelers, so they value a lot the sense of community of a place and its legal, bureaucratic, and flexible terms of living.

In line with the research aims, the paper first provides a detailed literature review, which was crucial for the development of the conceptual model and the research hypotheses. In Section 3, the study presents the methodological approach and the data collection process. Subsequently, the paper presents the results and discussion of the sustainable responsibility of digital nomads. Finally, Section 6 presents the theoretical and managerial implications of the results and offers suggestions for future research.

## 2. Literature Review

### 2.1. Digital Nomads

The emergence of the digital nomad was predicted by Makimoto and Manners (1997), who portrayed a future life simplified by portable technologies, in which people would be free to travel around the world while remaining connected to his or her job. In fact, the improvement of transportation systems and the unbundling of the tourism sector allowed the consumer to establish his own travel journey through online platforms, therefore facilitating the free movement and new mobile practices (Mancinelli 2020).

Digital nomadism is a location-independent lifestyle conducted, usually, by young professionals who work in an online basis, which allows them to travel and work simultaneously, blurring the lines between travel, leisure, work, and the boundaries between personal and professional life (Reichenberger 2018; Mancinelli 2020). By taking advantage of their spatial mobility and flexible working hours, and due to the lack of family commitments at earlier stages in life, digital nomads choose to explore the world (Reichenberger 2018; Richter and Richter and Richter 2020; Mancinelli 2020; Nash et al. 2021). Digital nomads are characterized as location-independent entrepreneurs or freelancers that are able to combine work and their personal life within high levels of flexibility (Müller 2016).

### 2.2. Hypotheses Development

Corporate social responsibility, hereon CSR, is one of the relationship development strategies which has become popular in the service industries around the world (Jeon et al. 2020). Customers are, also, becoming more concerned with company's behaviour and their external influence; therefore, CSR is often taken into consideration when making any purchase decisions (Castro-González et al. 2019). One important tool to fulfill a CSR approach is eco-innovation, consisting of a business strategy that encourages sustainability across a product's whole life cycle while simultaneously enhancing the productivity and competitiveness of an organization (Dias et al. 2021).

Social responsibility is a set of organizational actions, policies, and practices that ethically operate to contribute to an economic improvement along with engagement programs with the local community and its own employees (Jamali et al. 2015). It aims to provide wide social goods (Matten and Moon 2008; Varela and Dias 2015) and raise a just and sustainable society, with the help of its non-corrupt actions (Carroll and Shabana 2010). CSR is a business commitment whose actions should produce some social good beyond the interests of the company (McWilliams and Siegel 2001), and it has become an efficient concept for business strategy to positively impact society. It has recently emerged as a unique marketing tool for companies to create value and stable relationships with the consumers (Khan et al. 2015; Shah and Khan 2020).

Orlitzky (in Branco and Rodrigues (2006)) explains that CSR provides both internal and external benefits. Internally, it allows an efficient use of resources, lower costs, and the improvement of employee productivity. Externally, it enhances the reputation of a company, which is considered the competitive advantage of a firm. Therefore, to reduce negative environmental impacts, firms are changing their work relationships by increasing social and environmental awareness among employees or building a culture of volunteering by investing in local communities and in other stakeholders (Jamali et al. 2015). These ways of management and innovative practices are enabling companies to achieve a sustainable environment through eco-innovation as the primary objective. Eco-innovation is a part of the CSR activity to allow the customers to realize the positive performance (Mol 2003). As such, it is hypothesized:

**H1a.** *Social responsibility positively relates to entrepreneurial attitude.*

**H1b.** *Social responsibility positively relates to social entrepreneurial self-efficacy.*

**H1c.** *Social responsibility positively relates to entrepreneurial self-efficacy.*

**H1d.** *Social responsibility positively relates to eco-innovation.*

Drawing in the new ecological paradigm associated with the overall relationship between humans and the environment (Ntanos et al. 2019), we posit that environmental concern is described as an emotional reaction towards environmental issues, such as dislikes and compassion, from people that support efforts to solve environmental problems or have the willingness to contribute personally to their answer (Milfont and Gouveia 2006; Hu et al. 2010). This individual interest for environmental problems was treated as a relevant driver of environmentally conscious behavior, differing from energy conservation, waste recycling, and green buying behaviors (Hu et al. 2010; Manaktola and Jauhari 2007). A more social altruistic definition outlined environmental concern as a general mindset that reflects the extent to which the consumer is upset about threats to the environment, its consequences in the harmony of nature and future generations, and the lack of human action to react to these issues (Schultz 2001).

Additionally, Han et al. (2015) suggested that personal values could influence an individual's life for higher environmental concern. Moreover, Stern et al. (1993) opined a three-dimensional value orientation composed of egoistic, altruistic, and biosphere values that are significant when shaping a sustainable behavior. Therefore, the impact of values enhances an individual's environmental concern, norms, and attitudes that affect positively their environmental behavior (Choi et al. 2015). Along in line, it is also noticeable that

environmental concern significantly affects consumer's reactions towards eco-friendly products and services (Hartmann and Apaolaza-Ibáñez 2012), and later their attitudes and behavioral intentions (Kim and Han 2010). Data analysis reports that the basis of sustainability already runs like a continuous thread among digital nomads' lives and that the display of social, environmental, and cultural knowledge becomes the new token of this lifestyle. Therefore, the following hypothesis is proposed:

**H2a.** *Environmental concern positively relates to entrepreneurial attitude.*

**H2b.** *Environmental concern positively relates to social entrepreneurial self-efficacy.*

**H2c.** *Environmental concern positively relates to entrepreneurial self-efficacy.*

**H2d.** *Environmental concern positively relates to eco-innovation.*

Ajzen (2005) characterizes attitude as the impulse to proceed positively or negatively to an object, people, institution, or a moment. Therefore, entrepreneurial attitude can be described as an impulse to proceed positively or negatively to entrepreneurship. Entrepreneurs are people who have the capacity to manage and develop a new business by efficiently using resources to make profit and be succeed. As said before, CSR is a manifestation of good governance that, together with environmental concern, can affect a person's entrepreneurial attitude, as concluded by Indarti and Efni (2018). The authors indicated that a higher value for CSR funding was positively related with high levels of entrepreneurial attitude.

On the other hand, entrepreneurial attitude encourages the efficient distribution of natural resources and increases green practices, facilitating the integration of eco-innovation principles (Pacheco et al. 2010). At the same time, it affects human behavior through processes, goal setting, and outcome expectations, having, therefore, an impact in entrepreneurial self-efficacy (Bandura 2012). Therefore, it is hypothesized:

**H3a.** *Entrepreneurial attitude positively relates to social entrepreneurial self-efficacy.*

**H3b.** *Entrepreneurial attitude positively relates to entrepreneurial self-efficacy.*

**H3c.** *Entrepreneurial attitude positively relates to eco-innovation.*

Chen et al. (1998) explains entrepreneurial self-efficacy as the person's confidence in his/her capability to successfully accomplish the assignments required. These tasks reinforce business prospects, create original corporate settings, improve partners' relationships, help the company's significant objectives, adapt to outperform ecological troubles, and motivate workforce-gifted skills (Ahmed et al. 2021). People with an entrepreneurial mindset have more confidence in their competences and are less self-doubting, which contributes to innovative progress when confronted with difficulties and challenges (Lee et al. 2016), enabling firm performance.

Further, scholars have highlighted the role of self-efficacy as a variable in influencing individual behavior (Pihie and Bagheri 2010). Bandura (2012) verified that individual behavior is conceived by certain activities, such as the interaction of intrapersonal individuals' involvement and the circumstance. Interaction between these elements can format beliefs and influence behaviors (Pihie and Bagheri 2013). The point is that self-efficacy, by being seen as a social-cognitive process, is able to explain the impact of individuals' knowledge and actions in the form of attitude toward entrepreneurship. Self-efficacy notably influences the selection of human action despite the existence of alternatives, the volume of effort that it is spent to carry out the action, the perseverance in facing obstacles, and opportunities in acting (Pihie and Bagheri 2013; Shane and Delmar 2004). Similarly, Bandura (2012) concluded that self-efficacy is the main factor that affects behavior through the process, goal setting, outcome expectations, and challenges in the circumstances.

Therefore, Dwivedi and Weerawardena (2018) came up with a new but similar concept called social entrepreneurial self-efficacy that describes human behaviors that have influence on an individual's beliefs, efforts, levels of input, and persistence. It is viewed as

a strong predictor of self-confidence when facing uncertainty (Kakoudakis et al. 2017) and can be increased when interacting with external forces and the environment, thus moving towards value co-creation (Altinay et al. 2016). As it is said that most of digital nomads are entrepreneurs, it is hypothesized:

**H4a.** *Entrepreneurial attitude mediates the relation between social responsibility and social entrepreneurial self-efficacy.*

**H4b.** *Entrepreneurial attitude mediates the relation between social responsibility and entrepreneurial self-efficacy.*

**H5a.** *Entrepreneurial attitude mediates the relation between environmental concern and social entrepreneurial self-efficacy.*

**H5b.** *Entrepreneurial attitude mediates the relation between environmental concern and entrepreneurial self-efficacy.*

Eco-innovation can be described as a business method or production process that is new to the organization, which originates, throughout its life cycle, a decrease in environmental risk, pollution, and other negative impacts of resource use, when compared to other options (Kemp and Pearson 2007). Therefore, it refers to an innovation specifically focused on environmental impact (Bossle et al. 2016; Kiefer et al. 2017; He et al. 2018; Hojnik et al. 2018) whose new products use clean energy, are less polluting, and/or can be recycled, thus contributing positively to sustainability (Peng and Liu 2016; Severo et al. 2017).

Eco-innovation provides both environmental and economic advantages. For society in general, it shrinks the burden on the environment. For corporate businesses, eco-innovation enhances short- and long-term competitiveness and the creation of new markets. On the other hand, it builds or improves company reputation, but also decreases the costs, responds to new market demands, effectively fights intense competition, and complies with regulatory requirements (Sarkar 2013). Eco-innovation is a promising approach that decreases environmental impact and helps firms to increase their business value. As already noted, some digital nomads are entrepreneurs, therefore heavily focused on building or scaling up a business while engaging with the local communities or local projects. As such, it is hypothesized:

**H4c.** *Entrepreneurial attitude mediates the relation between social responsibility and eco-innovation.*

**H5c.** *Entrepreneurial attitude mediates the relation between environmental concern and eco-innovation.*

Figure 1 shows the conceptual model and hypotheses.

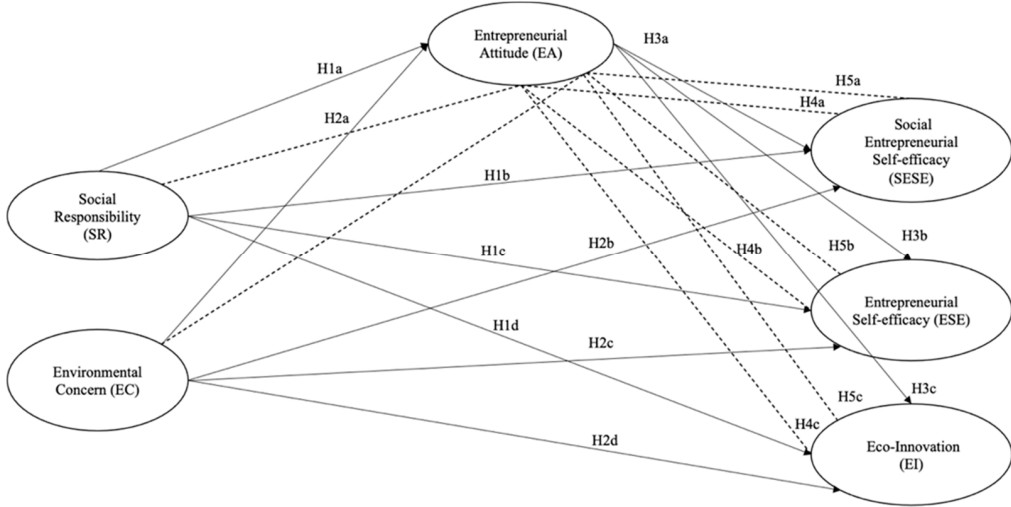

**Figure 1.** Conceptual model. **Note:** Dashed lines represent the indirect effects.

## 3. Materials and Methods

To study the sustainable responsibility of digital nomads, a quantitative method was used. The target population sample of digital nomads was recruited both via internet or at coworking and coliving spaces, using a snowballing technique. The questionnaire was delivered through an online platform and developed through a review of the literature. A pilot test was conducted in order to adjust some inaccuracies, and its final version took into consideration the suggestions made. A non-purposive convenient sample was adopted due to the lack of information about the total population sample. In the case of this study, the sample was obtained from digital nomads working from Portugal. The inclusion criteria are linked to the concept of digital nomads according to the Portuguese legislation: (i) working remotely with a labor contract, being a business owner or being a service provider (freelancer); and (ii) using mainly digital tools to perform the work. A total of 80 complete questionnaires were received between September 2021 and February 2022.

Of the respondents, 62.5% were female, and in terms of age, 21.9% were between 21 and 27 years old, 50% were between 28 and 34 years old, 12.5% were between 35 and 41 years old, 6.3% were between 42 and 48 years old, 3.1% were between 49 and 55 years old, and the remaining were older than 63 years old. Regarding the level of education, 43.8% had a bachelor's degree, 37.5% had a master's degree, and 9.4% completed high school. In terms of occupation, 46.9% were freelancers, 21.9% were running their own company, and 18.8% were full-time employees. Regarding the country of origin, 34% came from France, 32% from U.K., 12% from Brazil, 11% from U.S.A., and other small percentages from Canada and other European countries.

This study adopted existing scales to measure all variables. The social responsibility and the eco-innovation were measured using five items each that were adapted from Severo et al. (2018). Random items from the social responsibility variable are, "Whenever possible, before acquiring a product or service, I seek to know if the company has programs of engagement with the local community" and "I consider it fundamental to acquire products or services from companies that have an ethical, honest non-corrupt attitude". The four items used to measure the environmental concern were adapted from Verma et al. (2019), who originally adapted it from Abdul-Muhmin (2007), Cordano et al. (2011), and Kim and Choi (2005), such as, "The balance of nature is very gentle and can be easily upset" and "Human interferences with nature often produce disastrous consequences". The entrepreneurial self-efficacy and the entrepreneurial attitude were measured using a three- and a four-item scale respectively, adapted from Wardana et al. (2020). The social entrepreneurial self-efficacy was measured through a four-item scale adapted from Liu and Huang (2020). Random items from the social entrepreneurial self-efficacy variable are "I seek for new business opportunities for social change" and "I am creating new products/services to solve social problems". All the measures used a five-point Likert-type scale ranging from 1 = totally disagree to 5 = totally agree.

## 4. Results

To test the hypothesis model, structural equation modelling (SEM) was used. More precisely, a partial least squares (PLS) analysis was conducted, which is a variance-based structural equation modelling technique, by means of SmartPLS 3 software (Ringle et al. 2015). The results were analyzed and interpreted by a two-stage approach: first, an evaluation of the reliability and validity of the measurement model, and then the assessment of the structural model.

To determine the quality of the measurement model, the individual indicators of reliability, convergent validity, internal consistency reliability, and discriminant validity were examined (Hair et al. 2017). The results indicate that the standardized factor loadings of all items were above 0.6 (with a minimum value of 0.62) and were all significant at $p < 0.001$, which provided evidence for the individual indicator reliability (Hair et al. 2017). Internal consistency reliability was confirmed because all the constructs' Cronbach alphas and composite reliability (CR) values exceed the cut-off of 0.7 (see Table 1) (Hair et al. 2017).

**Table 1.** Composite reliability, average variance extracted, correlations, and discriminant validity checks.

| Latent Variables | $\alpha$ | CR | AVE | 1 | 2 | 3 | 4 | 5 | 6 |
|---|---|---|---|---|---|---|---|---|---|
| (1) Entrepreneurial self-efficacy | 0.700 | 0.811 | 0.590 | **0.768** | 0.802 | 0.343 | 0.104 | 0.228 | 0.645 |
| (2) Eco-Innovatio | 0.828 | 0.879 | 0.593 | 0.479 | **0.770** | 0.241 | 0.707 | 0.608 | 0.492 |
| (3) Entrepreneurial attitude | 0.912 | 0.938 | 0.792 | 0.396 | 0.136 | **0.890** | 0.489 | 0.533 | 0.502 |
| (4) Environmental Concern | 0.822 | 0.883 | 0.655 | 0.375 | 0.521 | 0.034 | **0.809** | 0.646 | 0.712 |
| (5) Social entrepreneurial self-efficacy | 0.922 | 0.945 | 0.812 | 0.569 | 0.485 | 0.658 | 0.307 | **0.901** | 0.474 |
| (6) Social Responsibility | 0.794 | 0.866 | 0.619 | 0.333 | 0.533 | 0.429 | 0.176 | 0.690 | **0.787** |

Note: $\alpha$—Cronbach Alpha; CR—composite reliability; AVE—average variance extracted. Bolded numbers represent the square roots of AVE. Beneath the diagonal elements are the correlations between the constructs. Above the diagonal elements are the HTMT ratios.

Convergent validity was also verified due to three key criterions. First, as illustrated before, all items loaded positively and significantly on their respective constructs. Second, all constructs had CR values higher than 0.70. Third, as Table 1 shows, the average variance extracted (AVE) for all constructs exceeded the threshold of 0.50 (Bagozzi and Yi 1988). The discriminant validity was evaluated using two procedures. First, the Fornell and Larcker criterion was used, which requires that a construct's square root of AVE (shown on the diagonal with bold values in Table 1) is larger than its biggest correlation with any construct (Fornell and Larcker 1981). Therefore, Table 1 shows that this criterion is satisfied for all constructs. Second, the heterotrait-monotrait ratio (HTMT) criterion was used (Hair et al. 2017; Henseler et al. 2015). As Table 1 shows, all HTMT ratios are below the more conservative threshold value of 0.85 (Hair et al. 2017; Henseler et al. 2015), which provides additional evidence of discriminant validity.

Finally, the structural model was assessed using the sign, magnitude, and significance of the structural path coefficients, the magnitude of the R2 value for each endogenous variable as a measure of the model's predictive accuracy, and the Stone Stone-Geisser's Q2 values as a measure of the model's predictive relevance (Hair et al. 2017). Nonetheless, the collinearity was tested before evaluating the structural model (Hair et al. 2017). The VIF values ranged from 1.032 to 1.266, which was below the indicative critical value of 5 (Hair et al. 2017), which indicated no collinearity. The coefficient of the determination R2 for the four endogenous variables of entrepreneurial attitude, social entrepreneurial self-efficacy, entrepreneurial self-efficacy, and eco-innovation were 18.6%, 68%, 30.2%, and 48%, respectively, therefore surpassing the threshold value of 10% (Falk and Miller 1992). The Q2 values for all endogenous variables (0.113, 0.512, 0.098, and 0.204, respectively) were above zero, which pointed out the predictive relevance of the model. We used bootstrapping with 5000 subsamples to evaluate the significance of the parameter estimates (Hair et al. 2017).

The results in Table 2 show that social responsibility has a significantly positive effect on entrepreneurial attitude (H1a), on social entrepreneurial self-efficacy (H1b), and on eco-innovation (H1d) ($\beta = 0.436$; $\beta = 0.457$; $\beta = 0.496$; $p < 0.05$). Additionally, environmental concern has a significantly positive relation with social entrepreneurial self-efficacy and with eco-innovation, providing support to H2b and H2d, respectively ($\beta = 0.211$; $\beta = 0.437$; $p < 0.05$). Entrepreneurial attitude has a significantly positive effect on social entrepreneurial self-efficacy (H3a) ($\beta = 0.455$; $p < 0.05$). Contrarily, all the other direct hypotheses weren't significantly positive because their $p$ values were above 0.05.

To test the mediation hypotheses (H4a–H5c), the propositions of Hair et al. (2017, p. 232) were kept. Therefore, a bootstrapping procedure was conducted to test the significance of the indirect effects via the mediator (Preacher and Hayes 2008), as shown in Table 3. The indirect effect of social responsibility on social entrepreneurial self-efficacy via the mediator of entrepreneurial attitude is significant ($\beta = 0.1986$; $p < 0.05$), therefore supporting H4a. Contrarily, all the other indirect hypotheses were not significant because the $p > 0.05$.

**Table 2.** Structural Model Assessment.

| Path | Path Coefficient | Standard Errors | *t* Statistics | *p* Value |
|---|---|---|---|---|
| Social Responsibility → Entrepreneurial attitude | 0.436 | 0.143 | 3.060 | 0.002 |
| Social Responsibility → Social Entrepreneurial self-efficacy | 0.457 | 0.163 | 2.811 | 0.005 |
| Social Responsibility → Entrepreneurial self-efficacy | 0.132 | 0.232 | 0.570 | 0.569 |
| Social Responsibility → Eco-Innovation | 0.496 | 0.171 | 2.909 | 0.004 |
| Environmental Concern → Entrepreneurial attitude | −0.043 | 0.162 | 0.263 | 0.793 |
| Environmental Concern → Social Entrepreneurial self-efficacy | 0.211 | 0.109 | 1.934 | 0.049 |
| Environmental Concern → Entrepreneurial self-efficacy | 0.341 | 0.231 | 1.473 | 0.141 |
| Environmental Concern → Eco-Innovation | 0.437 | 0.180 | 2.433 | 0.015 |
| Entrepreneurial attitude → Social Entrepreneurial self-efficacy | 0.455 | 0.137 | 3.318 | 0.001 |
| Entrepreneurial attitude → Entrepreneurial self-efficacy | 0.328 | 0.205 | 1.1597 | 0.111 |
| Entrepreneurial attitude → Eco-Innovation | −0.092 | 0.248 | 0.373 | 0.709 |

**Table 3.** Bootstrap results for indirect effects.

| Indirect Effect | Estimate | Standard Errors | *t* Statistics | *p* Value |
|---|---|---|---|---|
| Social Responsibility → Entrepreneurial attitude → Social Entrepreneurial self-efficacy | 0.199 | 0.097 | 2.054 | 0.041 |
| Social Responsibility → Entrepreneurial attitude → Entrepreneurial self-efficacy | 0.143 | 0.113 | 1.265 | 0.207 |
| Social Responsibility → Entrepreneurial attitude → Eco-Innovation | −0.040 | 0.0138 | 0.292 | 0.771 |
| Environmental Concern → Entrepreneurial attitude → Social Entrepreneurial self-efficacy | −0.019 | 0.078 | 0.249 | 0.804 |
| Environmental Concern → Entrepreneurial attitude → Entrepreneurial self-efficacy | −0.014 | 0.069 | 0.201 | 0.841 |
| Environmental Concern → Entrepreneurial attitude → Eco-Innovation | 0.004 | 0.043 | 0.091 | 0.927 |

## 5. Discussion

### 5.1. Social Responsibility: A Key Strategy

Customers are becoming very aware and interested in companies' behaviors and the influence they might have on their external environment. Hence, CSR is often taken into consideration when making any purchase decision (Castro-González et al. 2019), but it is also in the minds of entrepreneurs when starting a new business. This investigation validates the relation between social responsibility and entrepreneurial attitude, therefore empirically confirming the study of Indarti and Efni (2018), which indicates that high investments in CSR lead to high levels of entrepreneurial attitude. On the other hand, firms are changing their work environments and the relationships with their workers by increasing awareness about social and environmental issues (Jamali et al. 2015). This change in mentality has a positive influence in social entrepreneurship self-efficacy, as empirically confirmed in this study by the hypothesis H1b, corroborating the research from Altinay et al. (2016), which concludes that social entrepreneurial self-efficacy can increase when people interact with external factors and the environment.

Furthermore, this relationship can also be mediated by an entrepreneurial attitude. In fact, Lee et al. (2016) argued that people with an entrepreneurial mindset enhance firm performance due to their capabilities of easily overcoming challenges, which leads to high levels of self-confidence. This study corroborates the research conducted by Lee et al. (2016) by confirming hypothesis H4a. Additionally, this study confirmed the relationship between social responsibility and eco-innovation. Since this type of business method is characterized by using clean energy products that are less polluting, it contributes to high levels of sustainability (Peng and Liu 2016; Severo et al. 2017), improving the company's reputation (Pereira et al. 2021). Eco-innovation is the right approach for corporate social responsibility strategies because it helps firms to increase their business value while decreasing their environmental impact. Hence, the present study is empirically confirming the research developed by Sarkar (2013).

*5.2. The Role of Environmental Responsibility*

Environmental concern is a mindset affected and worried by the threats to the harmony of nature and future generations of the human species (Schultz 2001). These specific values, norms, and attitudes influence people's reactions towards eco-friendly products and services (Hartmann and Apaolaza-Ibáñez 2012), and later their attitudes and behavioral intentions (Kim and Han 2010). This study is empirically confirming these statements, since it concluded that there is a positive relation between the environmental concern and both social entrepreneurial self-efficacy and eco-innovation, H2b and H2d respectively. The influence of environmental responsibility has been studied in relation to other types of entrepreneurs, for example, in lifestyle entrepreneurs (O'Neill et al. 2022) or in small tourism firms (Dias et al. 2022), but not specifically to digital nomads, thus constituting a contribution of this study. Then, it is possible to assume that digital nomads, due to their interest in environmental issues and changes, adapt an eco-attitude in their daily lives by considering the products they consume and in having new business ideas that can help in those matters. On the other hand, the fact that this niche is highly composed of freelancers and young entrepreneurs, their entrepreneurial attitude, together with their preoccupation about the environment and social matters, will turn them into people aware of social inequalities with the eagerness to change them—which can be translated into high levels of social entrepreneurial self-efficacy, which corroborates hypothesis H3a.

## 6. Conclusions

This study contributes to the literature in several ways. First, it examines the relationship between several variables, having only digital nomads as the target niche. This allowed a deeper and different understanding about their lifestyle, the awareness of the best suitable environment for them, and a greater clarity about the relationship between the studied variables and the digital nomads, an underexplored theme until this point. This study was innovative in the context of digital nomads, especially in the identification of the multiple dimensions of social responsibility, which can be transformed into innovation and self-efficacy.

This information is particularly important when it comes to tourism facilities. Tourism businesses should understand that this type of traveler can be of huge importance due to their long-term stays and personal concerns. They are a different type of client, highly worried about the social responsibility of firms and the products they consume. If tourism companies start to truly worry about these matters and change their behavior, they could benefit from a community of travelers interested in the local community that will not harm the local environment and can improve the life conditions of the local community by generating profit in the local businesses and foment the economy. Furthermore, it was discovered that an entrepreneurial attitude could be a mediator between social responsibility and social entrepreneurial self-efficacy, a relationship that was insufficiently covered by the literature.

By conducting this research, the author contributed to the theory developed by Dunlap and van Liere (1978) and Dunlap et al. (2000) about the New Ecological Paradigm, in which broader issues, such as limits to grow and a steady-state economy, are taken into consideration when discussing environmental attitudes—also adapting the measuring scale created by the authors. Additionally, this study contributed to Theory of Planned Behavior developed by Ajzen (1995), considering that a person will successfully perform a behavior if he believes that the advantages of success outweigh the disadvantages of failure, having in mind internal and external factors.

It is possible to conclude that digital nomads are concerned about the social responsibility of the companies they work for, but also of hospitality providers, since it is where they spend much of their time. Policy makers and decision makers from destination management offices should take into consideration the impact they have in local communities and in the environment. Having an active role in those matters will always be a plus, both internally, where operations will be more efficient, but also externally by improving their

reputation and consequently increasing the number of satisfied clients. On the other hand, a big percentage of digital nomads are entrepreneurs, meaning that they own their own businesses. Due to their environmental and social concerns, they tend to create business ideas related to these matters, or at least that do not harm the local communities in any way. Therefore, policy makers and destination managers should pay attention to this fact and promote activities that foment sustainable responsibility and environmental concern, such as events, awareness actions, workshops, and fairs.

On the other hand, the destinations should facilitate the life of digital nomads in not only both legal and bureaucratic terms, but also in the sense of building a community for this specific niche. Most governments already offered a mix of "Nomad Visa" (that most nomads do not need), tax breaks, even though most nomads do not pay taxes in the country of residence, empty tourism pictures as promotion, and empty global promises on how good it is to work from there. However, nomads are looking for community, connection, giving back, and nature. Recently, Portugal approved a law proposal to create a Remote Work visa, seeing remote work with repopulation as a goal. In fact, it is important to help and empower the people that want to lead the change in small communities. It is necessary to give local government support to facilitate and reinforce the need for the leaders in the community to work together, to build dynamic coworking and meeting spaces that support the right activities, to bring people from the outside to inspire change and feed with inspiration, to understand that villages might be the best place to live, that cities are overcrowded and a person does not need to live in one to work in a big corporation, and finally, to understand that community is what humans seek and what brings the power to people. A very good example of what has to be made occurred on Madeira Island, where the most successful and original project focused on digital nomads in the world was conducted, a people-centric approach to a new reality where they provided the best experience by creating connections and the perfect conditions to work, live, and enjoy the islands.

This study contains limitations that indicate different paths for future research. First, the sample is small and limited to digital nomads that came to Portugal, and hence may not be generalized to other realities. Second, there was no consideration about cultural matters, which can influence a lot of the results of the sample. Therefore, it might be interesting to conduct comparative studies between different cultures. Another limitation which points for a future research direction is the need of comparison with other types of professionals and entrepreneurs, or, in other words, non-digital nomads.

**Author Contributions:** Conceptualization, Á.D. and I.M.; methodology, Á.D.; software, L.P.; validation, I.M. and L.P.; formal analysis, Á.D.; investigation, I.M.; resources, L.P.; data curation, I.M.; writing—original draft preparation, I.M.; writing—review and editing, Á.D. and L.P. All authors have read and agreed to the published version of the manuscript.

**Funding:** This research received no external funding.

**Institutional Review Board Statement:** Ethical review and approval were waived for this study since written informed consent was obtained for the in-depth interviews before each session. In the survey, a link to the online survey platform was sent by social media and partners' social media, and at no times was contact established between researchers and participants. Moreover, the interview script and the personal questionnaire did not include any information and on histories. As such, all data accessible to the researchers were stripped of respondents' names, addresses, or birth dates and cannot be linked back to them.

**Informed Consent Statement:** Informed consent was obtained from all subjects involved in the study.

**Data Availability Statement:** Data available upon reasonable request to the corresponding author.

**Conflicts of Interest:** The authors declare no conflict of interest.

### Appendix A. Questionnaire Items

| | |
|---|---|
| Social Responsibility | Whenever possible, before acquiring a product or service, I seek to know if the company has programs of engagement with the local community. |
| | I consider it fundamental to acquire products or services from companies that have an ethical, honest non-corrupt attitude. |
| | Whenever possible, before purchasing a product or service, I seek to know if the company has health safety actions to improve the quality of life of its employees. |
| | I consider it fundamental to acquire products or services from companies that do not use child labor and unfair remuneration. |
| | I value companies that respect equal pay for men and women. |
| Eco Innovation | I value companies that develop new recyclable or reusable products. |
| | I value companies that develop new products or services with the use of clean energy. |
| | Whenever possible, I try to buy innovative products that have low power consumption. |
| | I consider it important to purchase new products that are less polluting. |
| | I consider it important that new products reduce the environmental impact. |
| Environmental concern (EC) | The balance of nature is very gentle and can be easily upset. |
| | Human beings are severely abusing the environment. |
| | Humans must maintain the balance with nature to survive. |
| | Human interferences with nature often produce disastrous consequences. |
| Entrepreneurial Self-Efficacy | I could think creatively. |
| | I have an ability to commercialize new ideas. |
| | I have an ability to identify business opportunities. |
| Entrepreneurial Attitude | Career choice as an entrepreneur is interesting for me. |
| | Among the numerous choices, I would rather be an entrepreneur. |
| | Being an entrepreneur will give me extraordinary satisfaction. |
| | If I have opportunities and resources, I would like to start a business. |
| Social entrepreneurial self-efficacy | I seek for new business opportunities for social change. |
| | I am creating new products/services to solve social problems. |
| | I think creatively to benefit others. |
| | I think in commercializing an idea for social enterprise. |

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
