# Peer review of "Estimating the Impact of Digital Nomads’ Sustainable Responsibility on Entrepreneurial Self-Efficacy"

_socsci, doi:10.3390/socsci12020097_

Round 1

Reviewer 1 Report

General:

The aim of the article is to understand the sustainable responsibility of the digital nomads, to further conclude how they influence or could influence the local communities and how the tourism businesses can take an important step by understanding their needs and actions [Lines: 51-54 ].

The authors pose a research question - what are the constraints of the social responsibility of digital nomads? [Lines: 58-59]

A survey was conducted using an Internet platform. The questionnaires are based on a literature review [Lins: 245-246]

A non-purposive convenient sample of 80 respondents was adopted using the Internet, co-working spaces and co-living spaces using the snowball technique [Linse: 244-245; 246-250].

Six variables were used: social responsibility (5 items scale), eco-innovation (5 items scale), environmental concern (4 items scale), entrepreneurial self-efficiency (3 items scale), entrepreneurial attitudes (4 items scale), social entrepreneurial self -efficiency (4 items scale) [Lines: 258-273].

As part of the measurement, a five-point Likert scale was used [Lines: 273-274].

Based on the literature review, 17 hypotheses were put forward [Lines: 123-126; 151-154; 171-173; 204-211; 232-235].

In order to verify the hypotheses, structural equation modeling (SEM) was used, based on partial least squares (PLS) analysis, using the SmartPLS 3 software [Lines: 278-280].

Detailed comments:

[Line 186] Unnecessary paragraph in the middle of a sentence

[Line: 300] correct "constructs" to "constructs"

Summary:

It should be noted that the study was described in a very general way. There is no information about the geographical area of the surveyed people (from the last paragraph it can be concluded that these are people who came to Portugal). The question is whether it is appropriate to study such diverse groups as freelancers, people running their own business and full-time employees. Final conclusions are drawn on the basis of all data, and often refer only to individual subgroups. The question of how the sample was selected, the mere indication that the questionnaires provided by the online platform is insufficient. There is also no information in which sector of the economy they work and where they come from (domestic or foreign).

There is also no detailed information on the questionnaire itself. Only information on the number of items taken into account for a given variable is provided. The questionnaire should be attached as an annex.

For most of the work, only the relationships between the analyzed variables are analyzed in isolation from the characteristics of the tested sample. Then, far-reaching conclusions are drawn that are not directly related to the study, e.g. in relation to the tourism industry.

The question is whether the obtained results differ to any extent from the results of people who are not digital nomads.

In fact, the conducted study does not fully meet the indicated objective, as the interrelationships between variables are examined rather for a diverse and small sample.

The article requires significant changes.

Author Response

The aim of the article is to understand the sustainable responsibility of the digital nomads, to further conclude how they influence or could influence the local communities and how the tourism businesses can take an important step by understanding their needs and actions [Lines: 51-54 ].

The authors pose a research question - what are the constraints of the social responsibility of digital nomads? [Lines: 58-59]

A survey was conducted using an Internet platform. The questionnaires are based on a literature review [Lins: 245-246]

A non-purposive convenient sample of 80 respondents was adopted using the Internet, co-working spaces and co-living spaces using the snowball technique [Linse: 244-245; 246-250].

Six variables were used: social responsibility (5 items scale), eco-innovation (5 items scale), environmental concern (4 items scale), entrepreneurial self-efficiency (3 items scale), entrepreneurial attitudes (4 items scale), social entrepreneurial self -efficiency (4 items scale) [Lines: 258-273].

As part of the measurement, a five-point Likert scale was used [Lines: 273-274].

Based on the literature review, 17 hypotheses were put forward [Lines: 123-126; 151-154; 171-173; 204-211; 232-235].

In order to verify the hypotheses, structural equation modeling (SEM) was used, based on partial least squares (PLS) analysis, using the SmartPLS 3 software [Lines: 278-280].

R: we are thankful to the reviewer by all these highlights and for all the sharp identification of all the key issues in the paper.

Detailed comments:

[Line 186] Unnecessary paragraph in the middle of a sentence

R: Thank you for pointing this out. This mistake is now corrected.

[Line: 300] correct "constructs" to "constructs"

R: Thank you for pointing this out. This mistake is now corrected.

Summary:

It should be noted that the study was described in a very general way. There is no information about the geographical area of the surveyed people (from the last paragraph it can be concluded that these are people who came to Portugal). The question is whether it is appropriate to study such diverse groups as freelancers, people running their own business and full-time employees. Final conclusions are drawn on the basis of all data, and often refer only to individual subgroups. The question of how the sample was selected, the mere indication that the questionnaires provided by the online platform is insufficient. There is also no information in which sector of the economy they work and where they come from (domestic or foreign).

R: we agree that this information is important. As such, we conducted several changes in the methodology. First, we mentioned that the sample of taken from DN working in Portugal. We also added the inclusion criteria. Additional information is now provided regarding how the respondents were recruited and their country of origin. Please see the 1st and 2nd paragraph of the methodology.

There is also no detailed information on the questionnaire itself. Only information on the number of items taken into account for a given variable is provided. The questionnaire should be attached as an annex.

R: We agree with the reviewer. We now added the questionnaire items in the appendix.

For most of the work, only the relationships between the analyzed variables are analyzed in isolation from the characteristics of the tested sample. Then, far-reaching conclusions are drawn that are not directly related to the study, e.g. in relation to the tourism industry.

R: Thank you for pointing this out. To address this important recommendation, we revised the discussion section to align our results with previous research in tourism. We also provide a better link to the industry in the policy-making part of the conclusions.

The question is whether the obtained results differ to any extent from the results of people who are not digital nomads. In fact, the conducted study does not fully meet the indicated objective, as the interrelationships between variables are examined rather for a diverse and small sample.

R: Thank you for this important and sharp comment. In fact, it would be very interesting to compare with non-digital nomads and explore the differences. However, the inclusion criteria for our sample were focused on digital nomads, so we cannot provide this comparison with the existing sample. But we totally agree with the reviewer regarding the future exploration of this issue, as such, we added a mention to this in the future research part of the conclusions.

Reviewer 2 Report

The article presents a well-defined and current approach, adequately systematizing the methodology according to hypotheses supported by a broad theoretical literature.

However, the following aspects should be reviewed:

1. Perhaps it would be convenient to delimit in the title the geographical space (Portugal) where the research has been carried out, as indicated at the end of the conclusions. That way the content would not be so generic.

2. Perhaps, in the same sense, it would be useful to include the word "approximation" in the title. The number of surveys (80) is reduced to extrapolate and generalize the results, as the author also points out when referring to future research.

3. To complete the qualitative information, within the hypotheses formulated in relation to the variables of entrepreneurship, social responsibility, environmental concern, self-efficacy and eco-innovation, in addition to the theoretical compilation, it would have been interesting to add a contextualization through the information on the measures taken specifically in Portugal to attract digital nomads.

4. Sections 4.2 and 4.3 share the same title. Is it a bug? They probably refer to different concepts.

Author Response

The article presents a well-defined and current approach, adequately systematizing the methodology according to hypotheses supported by a broad theoretical literature.

R: Thank you for the supportive comment.

However, the following aspects should be reviewed:

  1. Perhaps it would be convenient to delimit in the title the geographical space (Portugal) where the research has been carried out, as indicated at the end of the conclusions. That way the content would not be so generic.

R: We agree with the reviewer. We extended the methodology to provide a geographical delimitation, in this case to DN in Portugal, and to describe their country of origin. We also revised the conclusions to provide a better link to the countries policy making.

  1. Perhaps, in the same sense, it would be useful to include the word "approximation" in the title. The number of surveys (80) is reduced to extrapolate and generalize the results, as the author also points out when referring to future research.

R: The reviewer poses an important issue. In order to address this recommendation we search for a synonym of approximation and we suggest the use of the term estimating. As such the new tile proposal is “Estimating the impact of Digital Nomads’ Sustainable Respon-sibility on Entrepreneurial Self-Efficacy”

  1. To complete the qualitative information, within the hypotheses formulated in relation to the variables of entrepreneurship, social responsibility, environmental concern, self-efficacy and eco-innovation, in addition to the theoretical compilation, it would have been interesting to add a contextualization through the information on the measures taken specifically in Portugal to attract digital nomads.

R: Thank you for pointing this out. In response to the reviewer recommendation we added additional information regarding the measures, including a full description of the items in the appendix.

  1. Sections 4.2 and 4.3 share the same title. Is it a bug? They probably refer to different concepts.

R: Thank you for pointing this out. We changed this mistake. The correct title is “5.2. The role of environmental responsibility”

Reviewer 3 Report

The topic is interesting, but there are some shortcomings as follows:

The topis and article is interesting. There are minor remarks from my side. They may enrich the paper should they be addressed. 

1. Abstract: The authors should update the abstract. They can remove a few sentences in the abstract relevant to the results and connect it with the article's title.

2. The Authors should explain Terms used in defined hypotheses (e.g., eco-innovation) before using them in hypotheses. 

3. Line 183-186- formatting errors

4. What is the research gap? What is/are the critical problem(s) facing digital nomads?

5. What were the criteria for assessing that the respondent is a digital nomad

6. The authors wrote that this research contributed to the theory e.g., New Ecological Paradigm. This theory should be introduced in the literature review. 

7. Please also elaborate if the current study is also consistent with findings from past studies in other country settings.

8. Please check the minor grammatical errors during the revision. Also, the first time, please use an abbreviation in parentheses. For example, please write the abbrevation of CSR.

9. The authors can also refer to the countries` policies to explain their results. 10. Also, the authors can explain with examples how the proposed method can be helpful for the governments in light of the actual situation.

11. The results of research and verification of hypothesis should be added. 

Point 4.2. Social responsibility: a key strategy and 4.3. Social responsibility: a key strategy – have got the same title. It should be changed.

12. The discussion should be added. 

Author Response

The topic is interesting, but there are some shortcomings as follows:

The topis and article is interesting. There are minor remarks from my side. They may enrich the paper should they be addressed. 

R: Thank you for the supportive comment and for the valuable recommendations.

  1. Abstract: The authors should update the abstract. They can remove a few sentences in the abstract relevant to the results and connect it with the article's title.

R: we agree with the reviewer. The abstract was revised for a more clear description of the results regarding social self-efficacy and self-efficacy, thus providing a better link to the title.

  1. The Authors should explain Terms used in defined hypotheses (e.g., eco-innovation) before using them in hypotheses. 

R: We agree with the reviewer. Yhe comcept of eco-innovation is now added to the literature review. Please see the 2nd paragraph of section 2.2.

  1. Line 183-186- formatting errors

R: Thank you for pointing this out. The error is now corrected.

  1. What is the research gap? What is/are the critical problem(s) facing digital nomads?

R: Thank you for pointing out this issue. We now present the research gap at the end of the 2nd paragraph of the introduction.

  1. What were the criteria for assessing that the respondent is a digital nomad

R: Thank you for alerting to this important issue. We now added the inclusion criteria for a DN to be considered in the sample.

  1. The authors wrote that this research contributed to the theory e.g., New Ecological Paradigm. This theory should be introduced in the literature review. 

R: Thank you for this important recommendation. We now introduced this theory to the literature review.

  1. Please also elaborate if the current study is also consistent with findings from past studies in other country settings.

R: We agree with the reviewer. We now extended the discussion for a more explicit presentation of the consistency regarding previous research or to describe where the study is advancing existing knowledge on the field.

  1. Please check the minor grammatical errors during the revision. Also, the first time, please use an abbreviation in parentheses. For example, please write the abbrevation of CSR.

R: Thank you for pointing this out. We now provide the full meaning before the abbreviation.

  1. The authors can also refer to the countries` policies to explain their results.
  2. Also, the authors can explain with examples how the proposed method can be helpful for the governments in light of the actual situation.

R: Thank you for these suggestions. We now provide a relation with the countries policies in the conclusions. We also expanded the conclusions to provide insights for policy making in the conclusions. Please see 4th paragraph of the conclusion.

  1. The results of research and verification of hypothesis should be added. 

R: we agree with the reviewer. Hypotheses testing is now presented in section 4.

Point 4.2. Social responsibility: a key strategy and 4.3. Social responsibility: a key strategy – have got the same title. It should be changed.

R: Thank you for indicating this mistake. The title is now correct.

  1. The discussion should be added. 

R: We now added a section 5 for the discussion.

Round 2

Reviewer 1 Report

General:

The understanding of the term 'digital nomad' within the meaning of Portuguese legislation has been clarified [lines: 256-259] and information on the research sample, i.e. the geographical area, has been supplemented [lines: 267-269]. The issues analyzed in the questionnaire are also included [lines: 646-647]. Additional limitations and the need for further research were also indicated [lines: 466-468]. Errors indicated in lines: 186 and 300 have been corrected. Minor additions have also been made and the structure of the paper has been improved.

Special issues:

[Line: 259] change comma to period

Summary:

Most of the reservations indicated in the review have been introduced or supplemented. It would still be worth discussing the questionnaire better, however, this is not an element disqualifying the paper. Some conclusions seem to go too far, however, this is a subjective opinion.